# Multi-angle pulse shape detection of scattered light in flow cytometry for label-free cell cycle classification

Daniel Kage [1], Kerstin Heinrich[1], Konrad v. Volkmann[2], Jenny Kirsch[1], Kristen Feher[3], Claudia Giesecke-Thiel [4,5,6✉] & Toralf Kaiser [1,6✉]

Flow cytometers are robust and ubiquitous tools of biomedical research, as they enable high-throughput fluorescence-based multi-parametric analysis and sorting of single cells. However, analysis is often constrained by the availability of detection reagents or functional changes of cells caused by fluorescent staining. Here, we introduce MAPS-FC (multi-angle pulse shape flow cytometry), an approach that measures angle- and time-resolved scattered light for high-throughput cell characterization to circumvent the constraints of conventional flow cytometry. In order to derive cell-specific properties from the acquired pulse shapes, we developed a data analysis procedure based on wavelet transform and k-means clustering. We analyzed cell cycle stages of Jurkat and HEK293 cells by MAPS-FC and were able to assign cells to the G1, S, and G2/M phases without the need for fluorescent labeling. The results were validated by DNA staining and by sorting and re-analysis of isolated G1, S, and G2/M populations. Our results demonstrate that MAPS-FC can be used to determine cell properties that are otherwise only accessible by invasive labeling. This approach is technically compatible with conventional flow cytometers and paves the way for label-free cell sorting.

[1] German Rheumatism Research Centre Berlin (DRFZ)—Flow Cytometry Core Facility, Charitéplatz 1 (Virchowweg 12), 10117 Berlin, Germany. [2] APE Angewandte Physik und Elektronik GmbH, Plauener Straße 163-165 / Haus N, 13053 Berlin, Germany. [3] EMBL Australia Node in Single Molecule Science, School of Medical Sciences, University of New South Wales, Sydney, Australia. [4] Flow Cytometry Facility, Max Planck Institute for Molecular Genetics, Ihnestraße 63-73, 14195 Berlin, Germany. [5] German Rheumatism Research Centre Berlin (DRFZ)—Cell Biology, Berlin, Germany. [6]These authors contributed equally: Claudia Giesecke-Thiel, Toralf Kaiser. ✉email: giesecke@molgen.mpg.de; kaiser@drfz.de

Flow cytometry (FC) is a powerful tool to investigate the physical and chemical characteristics of single cells and corresponding fluorescence-activated cell sorting is used to extract cell populations for further downstream analysis, such as gene expression by RNAseq. Flow-cytometric characterization is mainly based on the measurement of the intensities of scattered light and fluorescence light emitted from cells or particles after fluorescent labeling. For some purposes, such as cell cycle phase analysis, detrimental manipulation like fixation is required for DNA labeling, which restricts downstream experiments, or the fluorescent dyes are potentially toxic or interfere with cell functions[1,2] which may prevent the cultivation of cells after single-cell sorting. However, the identification and isolation of primary cell populations according to their cell cycle stages is of prime concern to, e.g., avoid that different cell cycle stages of cells introduce variance in single-cell RNAseq data[3] that would require correction in the data analysis. However, a flow-cytometric method to identify cell cycle stages without labeling is only available via elaborate analysis of imaging FC data[4] which is currently limited in throughput[5], or Raman-based methods[6,7] that focus on chemical composition but not on optical properties.

In addition to detecting fluorescence, a state-of-the-art flow cytometer measures the scattered light from cells. The scattered light signals are commonly detected at two angular ranges of ~0–20° (forward scattered light, FSC) and around 90° (side scattered light, SSC) relative to the direction of the excitation laser beam. The FSC correlates to the volume whereas the SSC signal correlates to the granular structure of the cell[8].

Since different scattering processes (diffraction, refraction, and reflection) dominate the intensity of scattered light at different angles[9], measuring light scattering over multiple angles provides more detailed information about the size, shape, morphology, and refractive index of particles[10,11] and cells[12,13]. For example, object size information is contained in small-angle scattering while shape information is found at larger angles[9]. According to Mie theory, variations in optical properties lead to angle-dependent intensity variations[14] which consequently suggest the angle-resolved detection. Previously, light scattering measurements with angular resolution in FC were used to distinguish infected and noninfected lymphocytes[15], to differentiate between normal and carcinoma cells[12], to differentiate cell lines[16], particles and cell subsets[17], or bacteria[18].

Flow-cytometric analysis of individual cells is achieved by passing the cells through a laser beam. Thus, in addition to the angle-dependent detection of scattered light, the scattered or fluorescence light intensity can also be measured as a function of time during the transit of the cell through the laser beam. Even though standard instruments do indeed acquire these intensity functions of time (pulse shapes (PS)), due to the technical setup, only three parameters (height H, area A, and width W) are derived from the PS in signal processing while the rest of the information is discarded. These key parameters enable the differentiation of cell doublets or tracking protein aggregation[19], for example. However, analyzing additional characteristics of the PS such as pulse skewness, kurtosis, or frequency properties, enables a more precise and specific differentiation of particles or cell doublets from large cell aggregates[20] as well as the differentiation of pollen, spores, and cell types[21]. Furthermore, PS measurements of fluorescence signals have been used in slit-scan cytometry to distinguish between chromosome profiles[22,23] or to distinguish between aggregates of cell nuclei[24]. Also in marine biology, PS detection has been used for the analysis of phytoplankton[25]. However, to the best of our knowledge, a combination of both angular and temporal resolution has not yet been investigated.

Here we report on the development, validation, and application of multi-angle (MA) light scatter detection in combination with time-resolved PS analysis for FC, combined in a method for cell cycle analysis hereafter referred to as MAPS-FC. For this purpose, we modified the optical setup of a commercially available flow cytometer, included a custom-built detector array for scattered light, and custom-made signal processing electronics. We further introduced a data analysis procedure based on wavelet transform and k-means clustering which facilitated the analysis of the PS data. In this proof of principle, the combination of hardware modification and data analysis allowed us to classify cell cycle subsets of HEK293 and Jurkat cells without the requirement of harmful cell labeling.

## Results

**PS characterization and cell morphology**. To determine the cell cycle phases of HEK293 cells, cells were stained with PI and BrdU and measured with MAPS-FC while also performing conventional fluorescence detection (Fig. 1a). Briefly, DNA intercalation of PI allows to determine the DNA content corresponding to DNA ploidy and to discriminate between G1, S, and G2/M phases of the cell cycle (Fig. 1b). BrdU allows a more specific distinction of S phase cells. Cell aggregates and debris were excluded by standard gating. When the dividing HEK293 cells are examined in an FSCM-SSC scatter plot (Fig. 1a), which is the analog of an FSC-SSC light scatter plot of a conventional flow cytometer, the G1, S, and G2/M populations were indistinguishable. Merely, a small increase of the FSCM signal with increasing cell division can be detected. PS from the detectors FSCL and FSCU are shown in Fig. 1c. The PS from HEK293 acquired in these two detectors showed differences in peak heights, ratios, and distances between the different cell cycle phases (Fig. 1c). For both, FSCL and FSCU, the peak distance increased with cell cycle progression by around 0.5 µs from G1 to G2/M. Furthermore, in FSCL the shape of the dip region between the two peaks was more structured in the S phase cells: while in G1 and G2/M it is mainly one dip with slightly varying width we observed plateau-like structures with oscillatory behavior in some S-phase cells. In the FSCU channel, the amplitude ratio of the two peaks increased from roughly 2.5 to 3 µs as cells progressed from G1 to G2/M. Cells in the S phase had a more homogeneous fall time of the first peak compared to G1 and G2/M cells (Fig. 1c). The PS of SSC signals did not differ much between the different cell cycle phases (not shown). Merely the height and width of the pulses slightly increased with progression through the cell cycle.

We next performed imaging FC analysis to image the HEK293 cells within each cell cycle phase, since the PS characteristics are considered a consequence of the morphological properties of cells progressing through the cell cycle. As expected, the cell size increased with progressing cell division (Fig. 1d). In addition, we examined the relative increase of the peak distances in the PS of FSCL and FSCU and compared these to the increase in cell diameter as obtained from imaging FC (Fig. 1e). We observed a correlation of all three curves with progression through the cell cycle. This suggests that the differences in the PS between the cell cycle phases shown in Fig. 1c result from morphological changes. We repeated these experiments with Jurkat cells and could confirm the results (Supporting Information Section 3.1, Figs. S3–4).

**Cell cycle analysis by MAPS-FC**. We next used wavelet transform followed by k-means clustering to analyze and group the PS acquired by MAPS-FC. The fluorescent staining was used as a reference to evaluate the performance of MAPS-FC. Extraction of features from the PS by wavelet transform has the advantage that specific features are conserved while unspecific parts are suppressed. For HEK293 cells, most information from the FSCL and

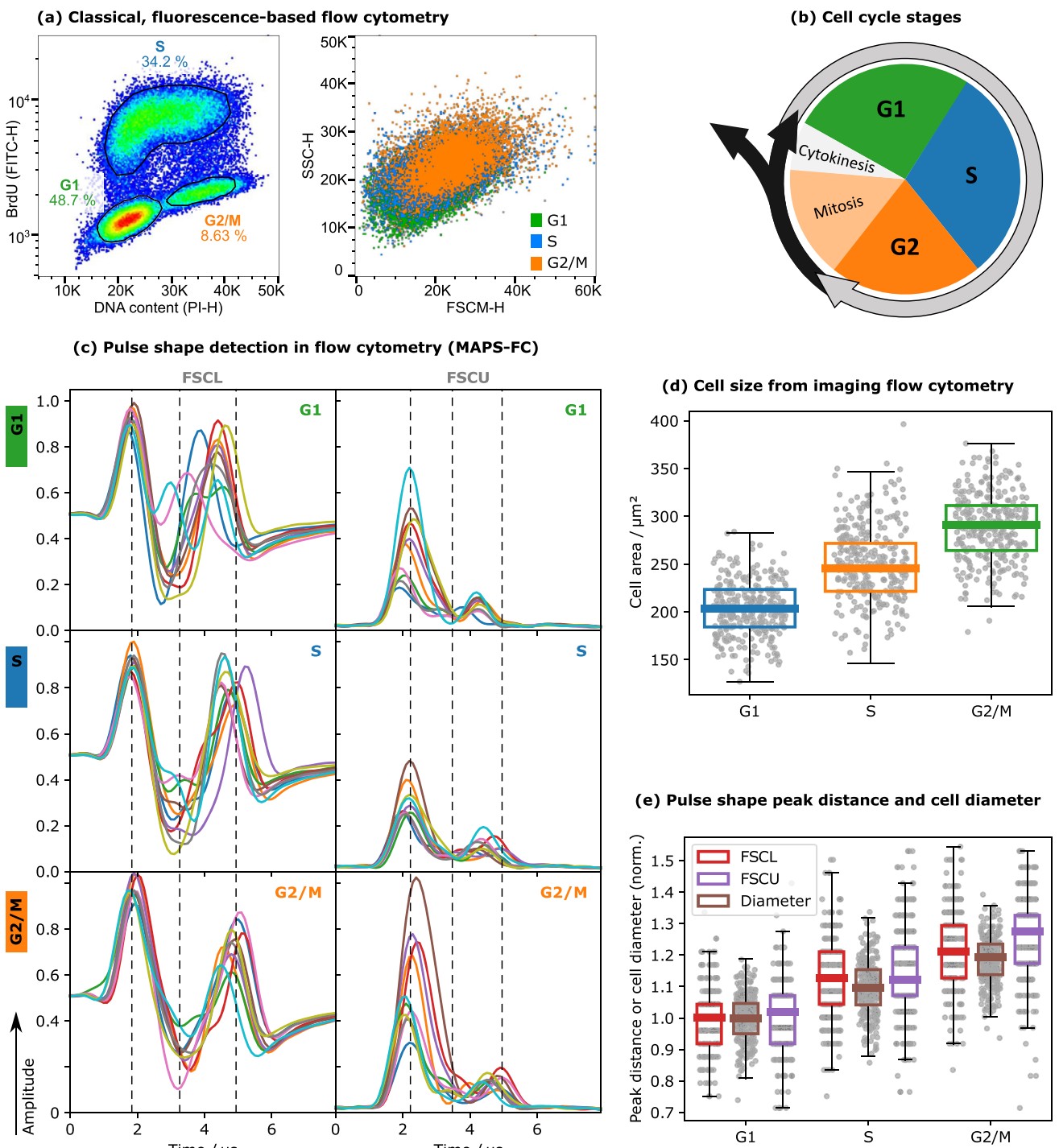

**Fig. 1 Cell cycle analysis by standard fluorescent staining, multi-angle pulse shape flow cytometry (MAPS-FC), and imaging flow cytometry.**
**a** Fluorescence intensity pseudocolor plot for PI and BrdU-FITC staining and corresponding FSCM-SSC dot plot. The populations are assigned to the phases of the cell cycle as indicated. Events were gated for single cells before (see Supporting Information Section 1). **b** Schematic representation of the cell cycle. **c** Representative pulse shapes in FSCL and FSCU for the three cell cycle phases G1, S, and G2/M. See Fig. 6 for details on the FSC channels. The vertical lines serve as guides to the eye for comparison of the PS. Each plot shows the PS, i.e., the scattered light intensity as a function of the travel time through the laser spot, for ten randomly chosen cells. **d** Mean cell size during cell cycle progression analyzed by imaging flow cytometry. Based on the area feature of the brightfield pictures, increasing cell size during cell cycle progression was determined ($n = 3061$ cells in G1 phase, $n = 1845$ cells in S phase, and $n = 484$ cells in G2/M phase, error bars indicate the upper/lower quartiles ±1.5 interquartile distances, 300 data points are shown with each box). **e** Relative increase of the mean peak distances in PS from FSCL and FSCU as well as the cell diameter throughout the cell cycle.

FSCU detectors was contained in wavelet decomposition level 4 (scale $2^4$) using the Haar wavelet. The clustering of the wavelet coefficients generated eight clusters per channel resulting in $8 \times 8 = 64$ combined clusters (an 8-by-8 contingency table) from the two detectors, FSCL and FSCU. Each cluster contained events with similar PS from both detectors. For an overview of the fluorescence intensity distribution in each of these clusters, Fig. 2a shows pseudocolor dot plots for all combined clusters. The contour lines in the background show the intensity distribution in all single cells for comparison. The numbers of the clusters are given for each subplot. For quantitative analysis, enrichment values were used to assign the combined clusters to the cell cycle phases (for details, see Supporting Information Section 2, Fig. S2), resulting in six groups: (1) G1, (2) predominantly G1 & S, (3) predominantly S & G1, (4) predominantly S & G2/M, (5) G2/M, and (6) undefined clusters. In Fig. 2a, five clusters are marked by a colored background. These five clusters that are specific and representative for five different stages of the cell cycle progression and their fluorescence intensity distribution is again shown in Fig. 2b along with additional details about the event distribution and enrichment of the clusters with cells from the three cell cycle phases. Many clusters spanned two adjacent cell cycle phases in terms of the staining with the majority of cells in one phase and a minority of cells in an adjacent phase as can be seen in Fig. 2a. This suggests that MAPS-FC resolves the continuous progression of cells through the cell cycle phases. Notably, clusters in the FSCL channel (horizontal direction in Fig. 2a) provide a coarse distinction between the cell cycle phases. Columns 2, 3, 5, 6, and 8 contain clusters with fluorescence properties indicating mostly G1 and some S phase cells. Columns 1, 4, and 7 contain clusters with cells in the G2/M and S phase. More detailed resolution and separation between G1, predominantly G1 and S, predominantly S and G2/M, and G2/M clusters is then provided by the FSCU channel (vertical direction in Fig. 2a). Importantly, for the G1 and G2/M phases, MAPS-FC clusters were found that exclusively contained cells with the respective staining pattern for only one phase at a time. This suggests that MAPS-FC can be used to identify G1 and G2/M phase cells and moreover detects fine variations during cell cycle progression. A comprehensive description of the assignment of all clusters to the cell cycle phases can be found in the Supporting Information Section 2, Fig. S2. Overall, 40 of 64 clusters were specifically assigned to phases of the cell cycle, comprising 78.4% of the single cells. An additional eight clusters were nonspecific to cell cycle phases and comprised 18.8% of cells, and 16 clusters were below the cluster size cutoff, representing 2.8% of events. We repeated this analysis with Jurkat cells (Supporting Information Section 3, Figs. S5–6). Also there, we found that most information for cell cycle phase classification was contained in wavelet decomposition level 4. We could confirm the resolving power provided by the FSCL and FSCU detectors, and again found clusters of exclusively G1 and G2/M phase cells, respectively (Supporting information Fig. S6).

**MAPS-FC analysis of pre-sorted G1, S, and G2/M cells**. To validate MAPS-FC, HEK293 cells were sorted for the cell cycle phases (G1, S, and G2/M) based on the fluorescent staining and subsequently analyzed with MAPS-FC. Figure 3a shows the fluorescence intensity distributions of an unsorted aliquot and the sorted aliquots measured with the cell sorter. For MAPS-FC on the sorted cells, k-means clustering was not performed again but instead, the cluster centroids of the unsorted sample were used and cells from the sorted aliquots were assigned to these cluster centroids. This allowed a direct comparison of the cluster sizes between the unsorted and the sorted aliquots. In the sorted populations, clusters assigned to the respective cell cycle phase should be enriched while other clusters should be depleted accordingly.

The clusters were grouped again into six groups as shown above (see Supporting Information Section 2 and Fig. 2 for details). The proportion of cells in the cluster groups was compared between the unsorted cells and the sorted cells (Fig. 3b). Clearly, the size of these groups changed in the sorted populations compared to the unsorted sample. The G1-sorted population exhibited an enrichment of the MAPS-FC cluster group assigned to G1 as well as the mixed clusters associated with G1 and S, and a depletion of the cluster group G2/M and the mixed clusters of S and G2/M cells. The aliquot sorted for the S phase showed an enrichment of the MAPS-FC groups associated with the S phase and overlap region with the G2/M, and depletion of pure G1, mixed G1 & S, and pure G2/M groups. Cells sorted for the G2/M phase displayed a clear enrichment of the G2/M-associated cluster groups and a depletion of the G1 and S groups. This clearly suggests that a correlation exists between the fluorescence-based sorting and the FSCL- and FSCU-based analysis by MAPS-FC. For each sorted aliquot, we determined the three clusters with the strongest enrichment. Figure 3c displays their fluorescence intensity distribution in the unsorted aliquot. Obviously, clusters that were enriched in the sorted populations show fluorescence intensity distributions that agree with the staining for the respective cell cycle phase which again supports that MAPS-FC identifies events with specificity for the cell cycle phase.

**MAPS-FC classification of live unstained cells**. Finally, we further tested and validated MAPS-FC by analysis of live HEK293 cells without DNA staining. MAPS-FC is based on the analysis of light scattered by cells and it is thus potentially influenced by cell fixation, as the reagents change the refractive index of the cells[26]. Cells were arrested in the G2 phase for 20 h and cell samples were analyzed with MAPS-FC at six time points 0 to 5 h after release from the cell cycle arrest and compared to an untreated sample (nc). An aliquot of cells from each time point was fixed and stained with PI to determine the cell cycle phases by conventional FC (Fig. 4a). As expected[27], the majority of cells in the sample measured immediately after release from the arrest (0 h) are found in the G2/M phase. With progressing time after release from the arrest, cells entered the G1 phase and the fraction of cells in the G2/M phase reduced. As time progressed further, some cells reentered the G2 phase adding up on other cells that had not yet excited from G2 synchronization. For MAPS-FC, the untreated control sample was used to define the wavelet-cluster centroids. Cells measured after cell cycle arrest were assigned to these predefined clusters to allow comparison. Since the purpose was to analyze unfixed cells, no staining was available for direct comparison. Assignment of the clusters to the cell cycle phases was therefore performed as follows: clusters that increased in size in the sample taken immediately after release from the arrest (0 h) with respect to the control sample are considered specific for G2/M. Vice versa, clusters that decreased in size upon G2 cell cycle arrest were considered as G1 & S phase clusters. Based on this assignment, the cluster sizes were compared with the results from the DNA-stained aliquots (Fig. 4b). The trends in sample composition were consistent between conventional FC and MAPS-FC analysis of unstained cells. These results indicate that MAPS-FC can be used to capture cell-specific cell cycle phase characteristics. In summary, MAPS-FC effectively determined cell properties that could be used to classify cells accordingly and moreover detected fine variations during cell cycle progression without the need of fluorescent staining.

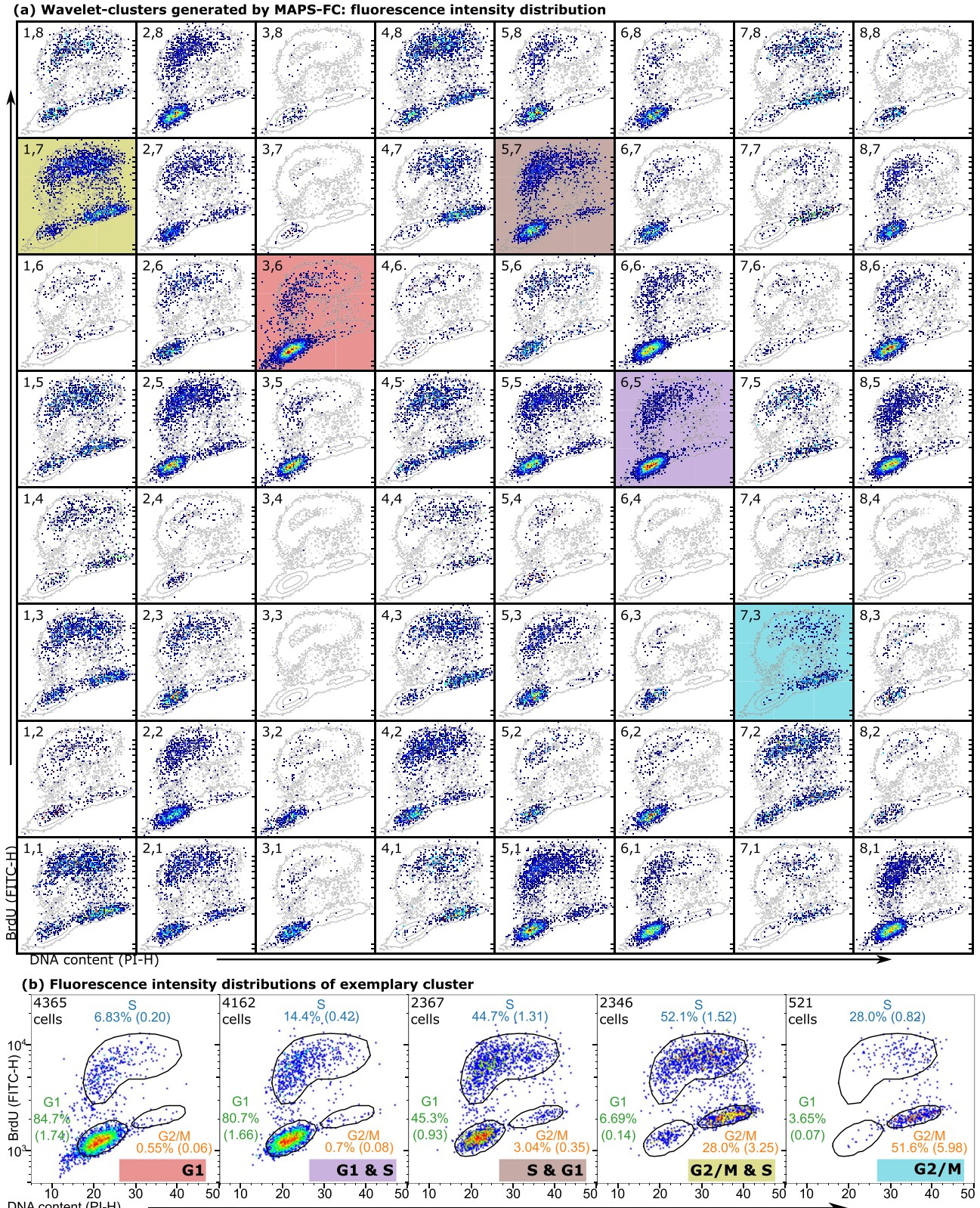

**Fig. 2 Cell cycle analysis with MAPS-FC. a** Pseudocolor dot plots of fluorescence intensity in each combined cluster. The contour lines show the intensity distribution of all single cells. Cluster numbers are given in each subplot. **b** Fluorescence intensity plots of exemplary clusters from panel (**a**) with further details. Values in parenthesis denote enrichment factor.

 

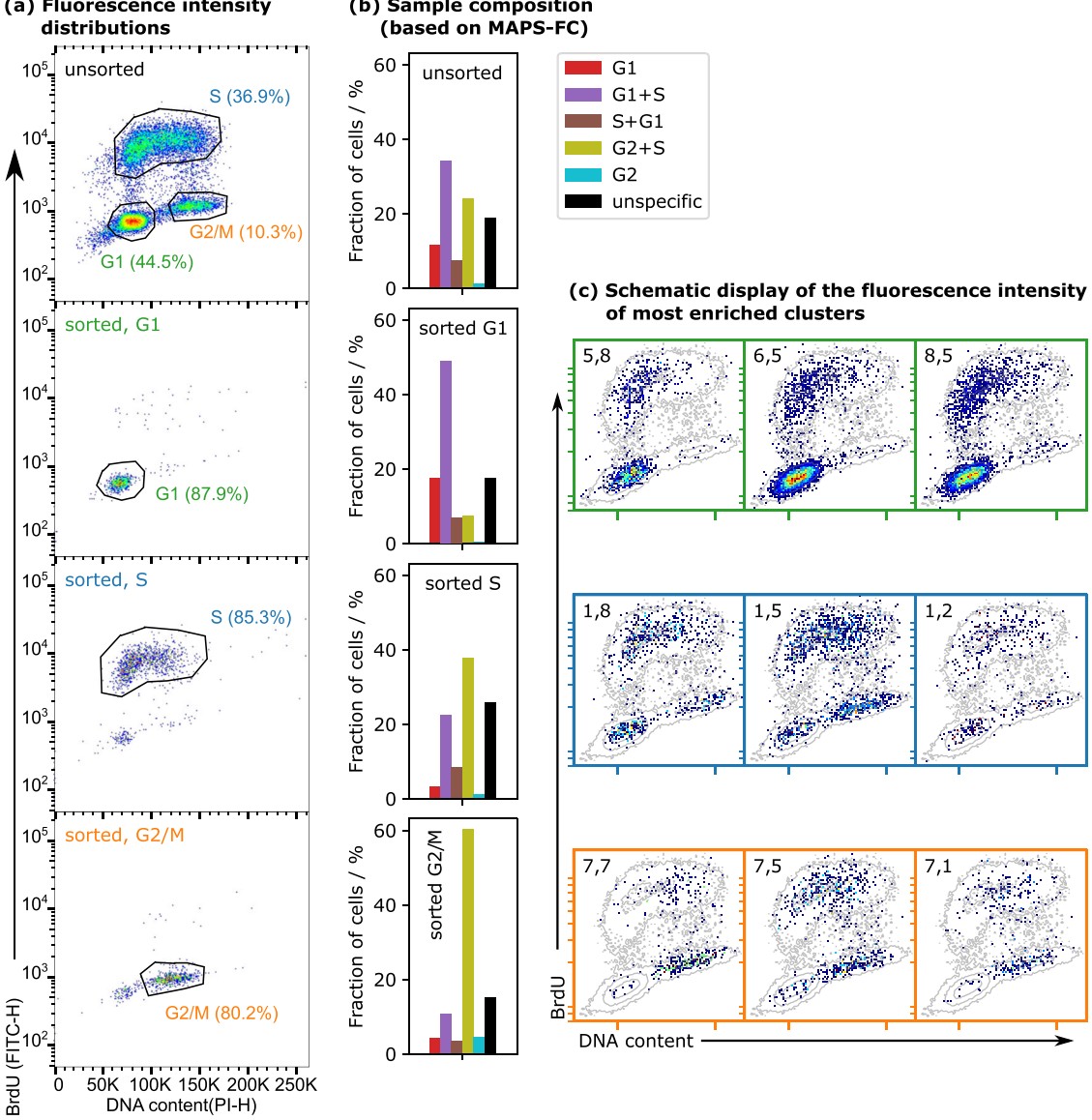

**Fig. 3 Sample composition of the unsorted sample and the aliquots sorted for the three cell cycle phases. a** Composition based on the fluorescent staining: sorted samples show a clear enrichment of cells in the respective cell cycle phase. The purity of the fluorescence-sorted cells was >80% without prior FSC-SSC gating. **b** Composition based on the wavelet-cluster analysis of unsorted and sorted G1, S, and G2/M populations. **c** Fluorescence intensity distribution in clusters in the unsorted sample that showed the strongest enrichment upon fluorescence-based cell sorting, i.e., a subset of Fig. 2a.

## Discussion

Here we have described a method (MAPS-FC) that enables the characterization of cells via the analysis of their time- and angle-resolved scattered light signals without the need of fluorescent staining. We used MAPS-FC to identify the cell cycle phases of HEK293 and Jurkat cells. Despite the different optical properties of the two cell lines and of fixed and unfixed cells, we demonstrated that MAPS-FC was able to reliably identify specific cell cycle subsets in all these settings. Our technique is compatible with standard flow cytometers in terms of data analysis, dynamic range, and cell throughput. It could be designed as an extension for existing instruments. Complex real-space imaging optics, femtosecond laser light sources, or log-in amplifiers as they are used in imaging, phase imaging[28], and Raman cytometry[6,7] are not required.

MAPS-FC extracts angle-dependent PS from the scattered light of cells. The characteristic light scattering stems from the optical properties of the cells and their changes during the cell cycle, like

morphology[21], size[29], or refractive index[30,31]. As a result of the data analysis performed, each cluster contained similar PS, and thus cells with similar intrinsic optical properties. This can lead to impure clusters with respect to cell cycle staining, but it groups cells in a way that is not accessible by fluorescent staining. Future downstream analysis of such clusters isolated by cell sorting based on MAPS-FC may provide insight into the nature of these cell cycle states and also more precisely specify clusters that we could not assign.

The PS obtained from low angles (FSCL) result mainly from light absorption. This measurement is comparable with the absorption or axial light loss[32] which provides information about the cell size and allows for a coarse distinction of G1 and early S phase cells from late S phase and G2/M phase cells. The differentiation of cells from G1 and early S phase or late S phase and G2/M was based on the PS from larger angles (FSCU), which result from the refractive properties of the cells and thus provide information about variations in the internal cell

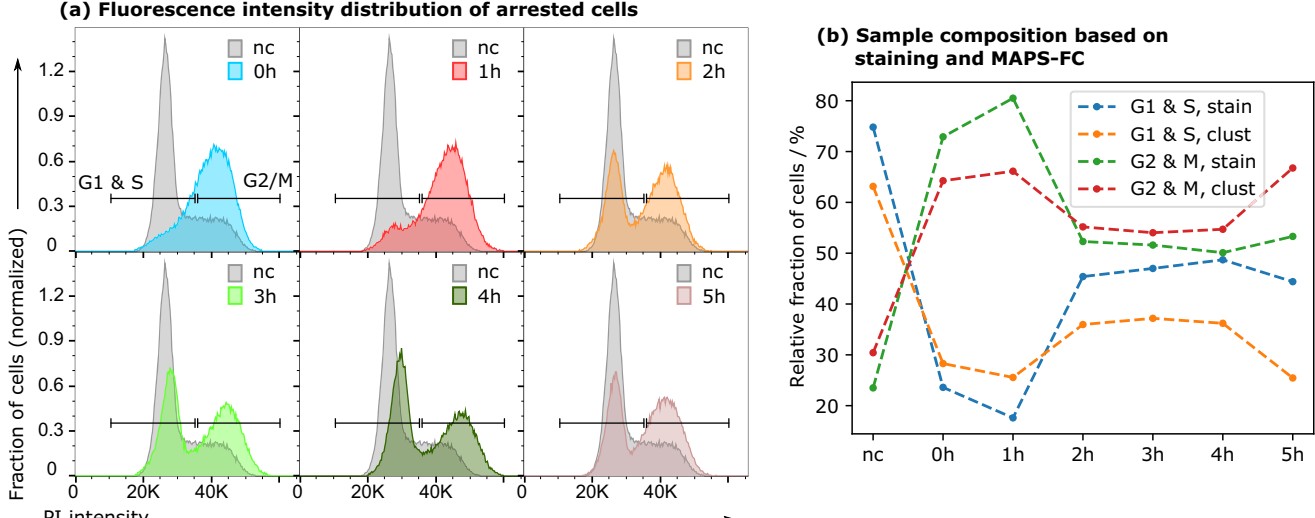

**Fig. 4 Analysis of sample composition by wavelet-clustering upon cell cycle arrest. a** Intensity histograms of PI staining in the samples measured at indicated time points after release from G2/M arrest, and the control sample (nc). To quantify the fraction of cells in G1 & S or G2/M phase, gates were defined as indicated in the histograms. **b** Sample composition at indicated time points. Orange and red curves show fractions of arrested but otherwise untouched cells in the respective cell cycle phase determined with MAPS-FC and cluster analysis. Blue and green curves show fractions of fixed and stained cells.

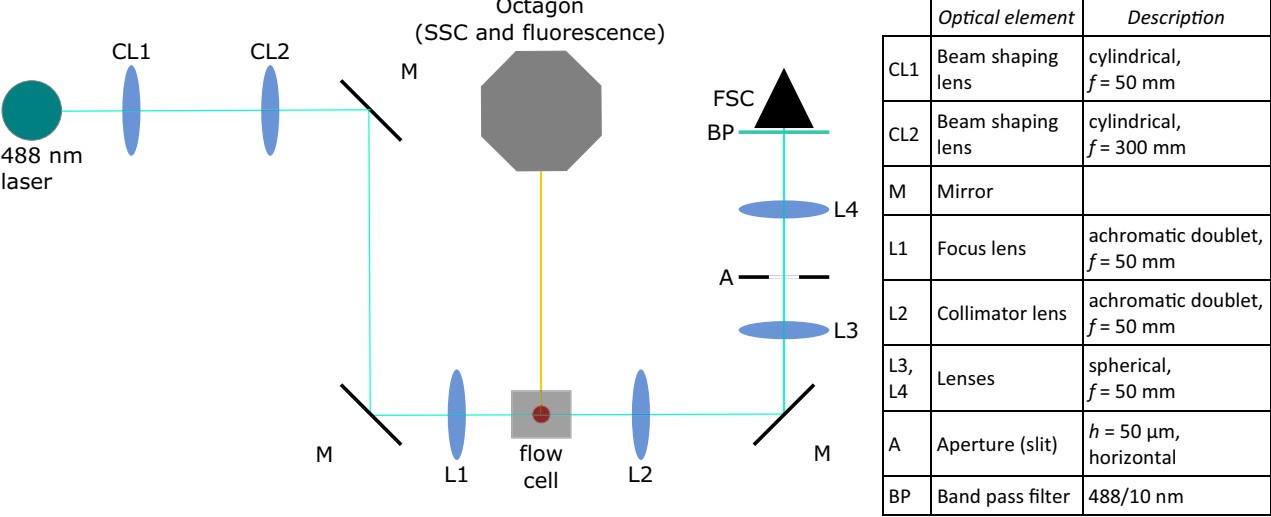

**Fig. 5 Schematic drawing of the setup of the modified LSR II.** The beam shaping optics were replaced with a telescope with cylindrical lenses (CL1, CL2) and a new focus lens (L1). The flow cell and side scatter (SSC)/fluorescence detection channel optics were kept from the original setup (Octagon). For the FSC channel, a collimating lens (L2) is followed by a spatial filter in an arrangement similar to a confocal microscope (L3, A, L4). FSC forward scatter, SSC side scatter.

structure, such as the position of the nucleus, which in turn induce variations in the local refractive properties of the cells[33]. Although the resulting clusters overlapped with respect to DNA staining, the results indicate that MAPS-FC enables further resolution of the continuous progression through the cell cycle. In fact, the conventional staining and MAPS-FC rely on different mechanisms: the fluorescent staining directly determines the DNA content[34] while MAPS-FC measures morphological changes. On the one hand, this hampers a quantitative comparison between the fluorescent staining and MAPS-FC. On the other hand, this finding is substantiated by the knowledge that the fluorescently labeled cell cycle populations are more heterogeneous than indicated by their mere DNA staining[35,36]. Thus, MAPS-FC can facilitate to study the continuous nature of the cell cycle progression. A detailed and well-founded

investigation of the subsets identified by MAPS-FC will, however, only be possible by cell sorting on PS characteristics, which will be implemented in future work.

MAPS-FC extracts features of the exact PS using wavelet transform. The subsequent $k$-means clustering is a vector quantization approach for grouping the PS according to their wavelet characteristics. Alternatively, multivariate methods such as PCA (principal component analysis), tSNE (t-distributed stochastic neighbor embedding), or artificial intelligence could be exploited for finding specific subsets by PS analysis. Moreover, the detector array presented here offers flexibility in the choice of 53 scattering angles. For the current measurements, we used a maximum of three angles of the array. However, for different cell types, a different set of angles or numbers might be beneficial and it could even be necessary to reconsider the laser beam geometry,

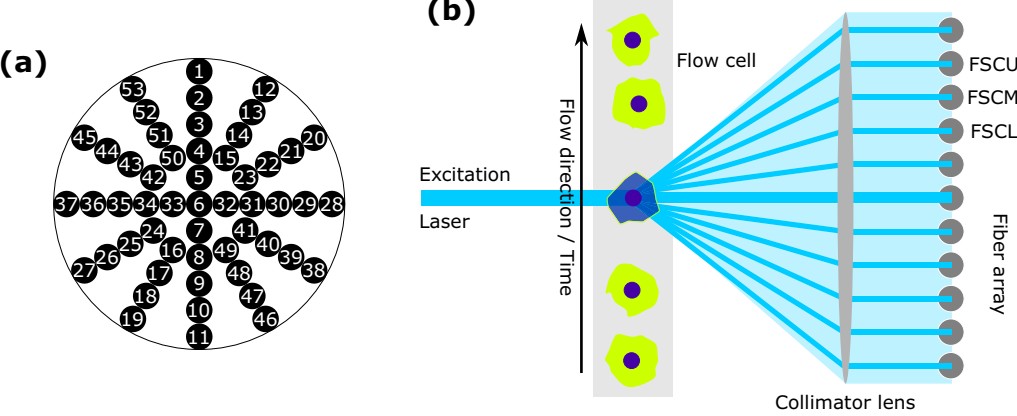

| Region | Center | Width | Fibers |
|---|---|---|---|
| center | 0° | 1.15° | 6 |
| first ring (4-point) | 1.72° | 1.14° | 5, 32, 7, 33 |
| second ring (first full 12) | 3.43° | 1.14° | **4**, 15, 23, 31, 41, 49, 8, 16, 24, 34, 42, 50 |
| third ring | 5.14° | 1.14° | **3**, 14, 22, 30, 40, 48, 9, 17, 25, 35, 43, 51 |
| fourth ring | 6.84° | 1.13° | **2**, 13, 21, 29, 39, 47, 10, 18, 26, 36, 44, 52 |
| fifth ring (outer) | 8.53° | 1.12° | 1, 12, 20, 28, 38, 46, 11, 19, 27, 37, 45, 53 |

**Fig. 6 Design of the FSC light detection system. a** Front view of fiber array. It consists of 53 individual fibers. Each fiber opening has a diameter of 1 mm and the distance between the fiber centers is 1.5 mm. **b** Schematic side view of the optical arrangement for FSC light detection with the fiber array providing angular resolution. For the sake of simplicity, not all optical elements are shown (lenses, aperture). The fiber array is located to the right and this side view shows the vertical row (1–11) in panel (**a**). Exemplarily, three detectors are labeled (exact positions on the fiber array and resulting detection angles can be seen in the Table).

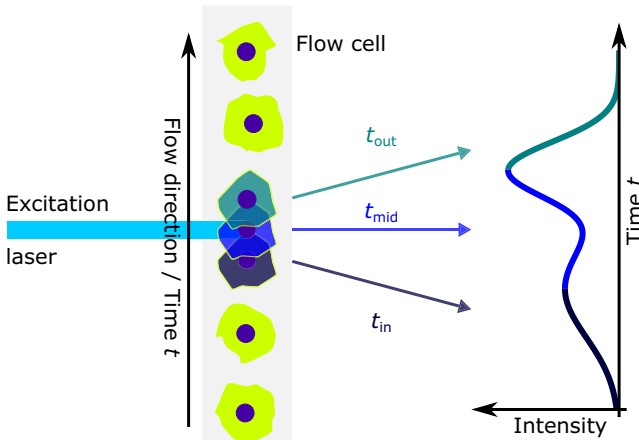

**Fig. 7 Illustration of pulse shape generation.** A cell moving through the laser beam creates a time-dependent intensity signal.

especially the spot size, depending on the size of the objects to be studied (e.g., bacteria or extracellular vesicles). Thus, alternative combinations of scattering angles need to be tested for their feasibility in cell analysis depending on the respective cell type. An important extension will be the future implementation of PS-based cell sorting. For the first prototype, the authors plan to use a piezo-driven mechanical sorting unit[37]. The sort decision has to be made within ~400 µs. A discrete wavelet transform and assignment to a cluster can be calculated on an FPGA board within this time span. This will allow for downstream analysis of subsets with various methods without using fluorescent staining.

Altogether, the proposed method complements the toolbox of state-of-the-art and evolving FC instrumentation[38] and contributes to a better understanding of the biophysical cell properties.

## Methods

**Cell preparation and analysis**. HEK293 cells (DSMZ) were grown in DMEM (Gibco) and Jurkat cells (DSMZ) were grown in RPMI 1640 medium (Gibco), supplied with 10% fetal bovine serum (FBS, Corning) and 2 mM L-Glutamine (Life technologies Ltd.) at 37 °C and 5% $CO_2$. To detect actively replicating cells, cells were seeded at 20–30% confluence and after 20 h of cultivation 60 µM Bromo-deoxyuridine (BrdU, Biolegend) was added for 1 h to the culture medium during conditions of exponential growth of the cells. Cells were then washed twice with 1x PBS (Th. Geyer) and harvested for fixation. Adherent cells (HEK293) were detached by trypsinization.

For cell cycle arrest, cells were seeded at 20–30% confluence and treated with 10 µM RO-3306 (Sigma) for 20 h. Cells were released from G2 arrest by washing twice with fresh medium for 3 min. Analysis of cell cycle stage was performed between 0 and 5 h after substitution of RO-3306 with fresh media.

For fixation, 4.5 mL of cold methanol/acetone (−20 °C) were added to $1 \times 10^7$ cells in 0.5 mL PBS in a drop-wise manner while vortexing. Cells were incubated overnight at −20 °C. After fixation, cells were centrifuged and washed once in PBS. Cells were then treated first with 2 M HCl/0.5% Triton X-100 for 30 min at room temperature (RT). After centrifugation, cells were neutralized by resuspending in $Na_2B_4O_7$ and incubated for 2 min at RT. Cells were washed once in PBS/1% BSA and resuspended in PBS/1% BSA/0.5% TWEEN20 and incubated with anti-BrdU-FITC antibody (1:25, Thermo Fisher Scientific, LOT 1982681) for 1 h. For DNA staining, cells were incubated with propidium iodide (PI) staining solution (PBS containing 50 µg/mL PI, 100 µg/mL RNase A, 2 mM $MgCl_2$) for 20 min at RT.

For validation of MAPS-FC, cell sorting was performed on a FACSAria II (BD) cell sorter. PI fluorescence was detected through a 670/30 band pass filter. Additionally, BrdU-FITC fluorescence was detected through a 530/30 band pass filter. Cell aggregates were excluded using a 670/30-width vs. 670/30-height plot. G1, S, and G2/M subsets were sorted based on PI and BrdU staining. The cells were divided into two aliquots. From one aliquot the G1, S, and G2/M phases were sorted and then analyzed by MAPS-FC. The other aliquot was analyzed by

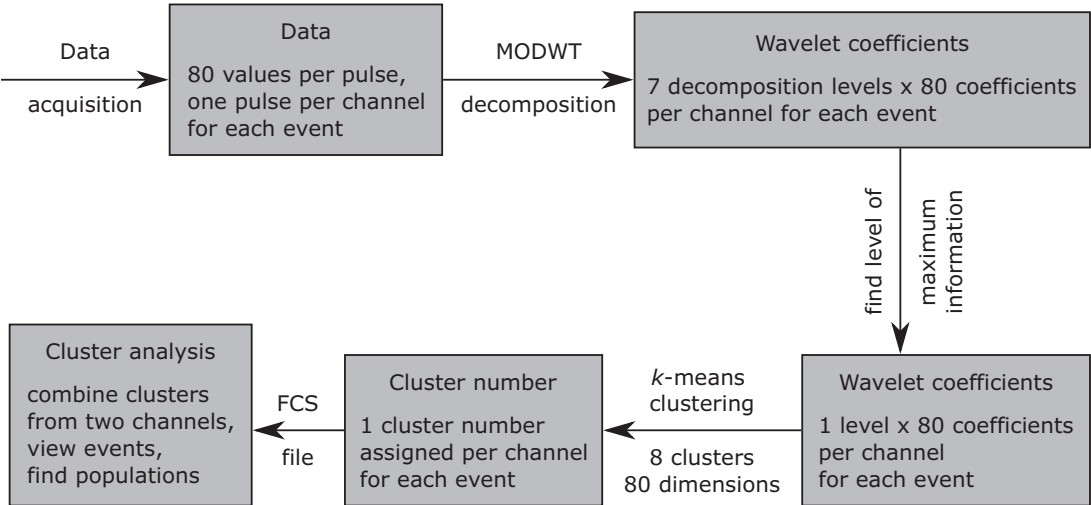

**Fig. 8 Flow chart of the data analysis procedure.** By maximum overlap discrete wavelet transform (MODWT), the relevant information of the PS is extracted. The *k*-means clustering reduces the data dimension from 80 data points down to one specific cluster number.

MAPS-FC without sorting. Both aliquots were stained since the cell cycle staining was used as a control for MAPS-FC.

For imaging FC, fixed and stained cells were measured on an Amnis® ImageStream® X MkII Imaging Flow Cytometer (Luminex) with 40× magnification controlled by INSPIRE software (Version 200.1.388.0) and fully ASSIST calibrated. Per sample, 20,000 events were acquired. Data analysis was performed using IDEAS 6.2 software.

**Optical setup for MAPS-FC.** The original optical setup of an LSR II (BD) was replaced by a new optical arrangement depicted in Fig. 5. The 488-nm laser, flow cell, fluidics, and fluorescence detection unit of the original LSR II were reused.

For excitation beam shaping, a telescope-like assembly of two cylindrical lenses (CL1 and 2) was used to create an elliptical beam. The beam is focused into the flow cell with the achromatic focus lens (L1), which creates a smaller beam waist than a standard lens. The focal spot is a horizontally stretched ellipse (short axis along flow direction) with a smaller spot size than in the original optics. The short axis is estimated to be around 5.5 to 6 µm (full width at half maximum, FWHM). Considering the typical size of eukaryotic cells, this situation is comparable to scanning FC with a tightly focused laser beam[39]. The laser power reaching the flow cell amounts to 12 mW. Fluorescence signals and SSC were detected with band pass filters for 530/30 nm (FITC), 676/29 nm (PI), and 488/10 nm (SSC) using the original LSR II detection optics. For FSC detection, the modified optical setup consists of three lenses L2-4 and a slit aperture A (perpendicular to the flow direction). The specifications of the optical elements are summarized in Fig. 5.

An array of 53 optical fibers connected to photomultiplier tubes (PMTs) was used for angle-resolved scattered light detection. A front view of this fiber array is shown in Fig. 6a. Each fiber opening has a diameter of 1 mm and the spacing between the fibers is 0.5 mm which enables an angular resolution of around 1°. A simplified side view of the FSC detection optics is displayed in Fig. 6b. For the cell cycle studies reported here, only three of 53 detectors labeled FSCL/M/U (lower/middle/upper) were used due to the limited number of signal processing channels (maximum 8). Fiber positions at the vertical line were chosen since more PS heterogeneity was expected along the flow direction and these three fibers detect high signal levels at low background. Details on the fiber arrangement can be found in the listing in Fig. 6. With respect to the fiber numbers listed in Fig. 6, the numbers of the three detectors used for the cell cycle analysis are 2, 3, and 4.

**Data acquisition/signal processing.** The original LSR II signal processing electronics were replaced by custom-made amplifiers and digitizers (APE)[40]. It provides eight detection channels with linear analog amplifiers suitable for photodiodes and PMTs. The SSC channel is chosen to be the trigger for data acquisition as it provides a common trigger point for the FSCL-U channels. As each object transits through the laser beam, a curve of scattered light intensity vs. time is created (PS), see Fig. 7. The signal processing electronics enable the PS to be recorded in each channel. This is not currently possible in commercially available cytometers as these instruments store list-mode data of derived parameters of the PS such as pulse height, transit time, and area.

The analog signals were digitized at a sampling rate of 10 MHz providing a time resolution of 0.1 µs. Unspecific high-frequency components were removed by a low-pass filter (7 MHz cutoff) integrated into the analog amplifiers. The PS were recorded within a time window of 8 µs (trigger point at 2 µs) and were stored in a

binary file format with 16-bit resolution. The flow speed is determined by the fluidics pressure. We used 4 psi, which results in a flow speed of about 5 m/s. The maximum length of the trigger window is 16 µs. Coincidental detection of multiple objects was avoided using low event rates (<1000 evts./s). Simultaneously, a standard FCS 3.0 format file with typical H, A, and W parameters for each event and detected light channel was generated. Both files can be connected such that for each event in the conventional FCS file the respective PS are available. This enables visualization by commercial data analysis software such as FlowJo (BD).

**PS data analysis.** The raw data contains one PS per event (cell or particle) and channel. Each PS contains 80 data points, which results in $8 \times 80 = 640$ data points for each event. Data analysis is performed off-line after the measurements. The pulse characteristics are extracted by a wavelet-based data decomposition. In short, a wavelet transform is a mathematical tool for digital signal analysis. Similarly to a Fourier transform, the signal is projected onto certain basis functions called mother wavelets. In contrast to the Fourier transform, the wavelet transform retains information on the localization of features and provides a decomposition of the signal on discrete scales into wavelet coefficients[41–43]. Here, the maximum overlap discrete wavelet transform (MODWT) was employed. Subsequently, the feature level carrying the relevant information was determined. To this end, the PS were reconstructed from the wavelet coefficients from only one feature level at a time. In each time sampling point, the standard deviation across all reconstructed pulses was inspected. The feature level where strong variations appear is considered carrying relevant information. The resulting wavelet coefficients were then clustered by *k*-means clustering in order to group PS with similar characteristics. In short, the wavelet transform of the signal, i.e., the 80 coefficients, is viewed as an 80-dimensional vector, and clusters are built in the corresponding 80-dimensional space. Each measured event is assigned to exactly one cluster out of eight clusters per channel (including all scatter and fluorescence channels). The procedure is summarized in Fig. 8. For the analysis of the events in the clusters, the clusters from two FSC channels (FSCL and FSCU) were combined in conjunction. This increases specificity and leads to $8 \times 8 = 64$ clusters combined in a contingency table. The events within each entry in the table were also analyzed for their fluorescent staining to assign clusters to the cell cycle phases for cluster interpretation. The PS from the FSCM channel are redundant to the FSCU channel and are therefore not used for further data analysis.

The described data analysis was performed with Matlab scripts (Matlab R2019b, The MathWorks) using the built-in Matlab functions for MODWT and *k*-means clustering with squared Euclidean distance metric. Third-party packages[44,45] were used for importing and exporting FCS list-mode files for further analysis in FlowJo v.10.6.2 (BD). Moreover, a custom-built FlowJo plugin (APE) was used to display the PS of the events within any gate of choice. This allowed to analyze connections between conventional data analysis and the PS.

**Statistics and reproducibility.** As indicated in the article, all measurements were performed at least in triplicates on two different cell lines. For each MAPS-FC measurement, $10^5$ events were recorded. At Amnis, 20,000 events were recorded. The MAPS-FC data analyses and the generation of the corresponding dot plots were performed using custom code in Matlab or Python. *K*-means clustering was used to group the PS. Euclidean distance was chosen for cluster distance determination. A special FlowJo plugin was used for data analysis of raw PS. The PS data (16 bit, 10 MHz) were recorded using custom-built hardware.

**Reporting summary**. Further information on research design is available in the Nature Research Reporting Summary linked to this article.

## Data availability
The datasets generated and analyzed during the current study are available from an open repository (https://doi.org/10.5281/zenodo.5235706)[46]. Any remaining information can be obtained from the corresponding author upon reasonable request.

## Code availability
The computer code used for the presented study is available from an open repository (https://doi.org/10.5281/zenodo.5235706)[46]. The code can be used freely without any restrictions under the MIT License. However, third-party software with restrictive terms of use is required for execution.

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

## Acknowledgements
We thank Sebastian Falkowski from APE for the development of the FlowJo plugin. The authors acknowledge funding by the Investitionsbank Berlin (IBB), grant 10165404.

## Author contributions
D.K., T.K. and K.H. conducted the flow cytometry measurements. K.H. and J.K. prepared cell samples. T.K., D.K. and K.v.V. planned and set up the modified optics and measurement electronics. D.K. and T.K. performed data analysis. T.K., C.GT. and K.F. developed the concept of pulse shape analysis. All authors were involved in data interpretation and discussion.

## Funding

## Competing interests
K.v.V. is employed by APE who provided the data acquisition electronics and might benefit as a company from the interest in the presented method. All other authors declare no competing interests.
