## [Peer Review File · Communications Biology]

Multi-angle pulse shape detection of scattered light in flow cytometry for label-free cell cycle classificationPoint-by-point response to the reviewers' comments

The authors greatly acknowledge the valuable feedback from the reviewers. We will address all aspects raised in the following point-by-point response.

Reviewer #1:

- In the late 1980s Galbraith et al posed the idea of using wavelet analysis with time-of-flight to characterize cells for separation of their type and function. Others in the early 1990s at Los Alamos generated a massive multi-angle cytometer to characterize cells with different angles and polarization. While the authors combine these two ideas, the combination thereof is not seen as a major step in the field. The approach would be difficult for the broad field, who do not have the novelty of analysis nor the ability to modify the data acquisition approaches and personal cytometry systems.

We agree with the reviewer's statement that the individual basic concepts of pulse shape and light scatter characterizing were published elsewhere (the work from the Los Alamos group is cited in our manuscript). However, to the best of our knowledge, wavelet analysis of flow cytometric pulse shapes has not been published previously. We would appreciate details regarding the work of Galbraith et.al. to use it in our future work. We would be very happy to support other researchers who want to use our technique in implementing the method with their setup. Documentation on hardware and algorithms is available.

- It is not clear if cells **should** be labeled for characterization owing to the uniqueness of absorption and how that influences intensity of Rayleigh light scatter. If so, would not this circumvent the ease of non-labeled samples?

We apologize for the imprecise wording. Extinction also appears for unstained cells. Fixation changes the optical properties of cells and consequently also the scattering properties. As demonstrated in our work by analyzing unstained cells, this does not affect the ability of MAPS-FC to characterize the cell cycle phases. The staining was merely used as a reference to evaluate the specificity of MAPS-FC. We clarified this point by explicitly mentioning the use of the staining for reference purposes in the Methods and Results sections. A sentence was added in line 85 in the original document (lines 87-88 in the revised manuscript).

- Are all analyses performed off-line? If so, how might this approach be adopted for real-time histogram formation or cell sorting?

The analysis is performed offline at the moment. We apologize for not mentioning this in our Methods Section and added the respective information. The signal processing can be adapted for online capability. We also provide a brief comment on this issue in our Discussion Section. A sentence was added in line 136 of the original document (line 145 in the revised manuscript).

- Line 28: This could be seen as an inaccurate statement considering the authors showed in Fig. 5 that you could probably separate cells by their position in the cell cycle using imaging flow cytometry and cell size analysis (though it would likely be less specific/accurate).

We apologize for the inaccuracy in this statement. Indeed, literature reports on image-based discrimination of unstained cells in different cell cycle phases exist. We added corresponding references and very briefly discuss the matter in the Introduction Section. The sentence starting in line 28 in the original document was modified (line 29-30 in the revised manuscript).

- Line 31: The statement of morphological complexity seems quite vague to the point of an incorrect statement. Might you then simply compare the range of this complexity with a simple cytometry in comparison to the new approach?

We apologize for the vagueness of our statement. We specified the sentence.

The sentence ending in line 31 in the original document was shortened. A more specific sentence with an additional reference was added (lines 33-34 in the revised manuscript).

- Lines 39-41: The author should elaborate on specific examples referenced here with respect to angular-resolved cytometry.

We disagree with the reviewer's comment, as we have briefly mentioned the purpose for which the various stray light measurements have already been used. For more details, the interested reader is referred to the cited publications.

- Line 44-46: This is not entirely true. Most commercial instruments may only report A,H,W, they include other processing steps that might not "appear" to the user. Many There are also examples of custom built instruments custom DAQs that have demonstrated this. Time-analysis (time-of-flight) has been incorporated into commercial systems for years.

We appreciate the comment and clarified that commercial instruments measure the pulse shape and calculate Height, Area, and Width from the pulse shapes. However, commercial instruments do not store and provide the detailed pulse shapes to the user. Moreover, the calculation of pulse area and width is based on closed-source internal signal processing algorithms which are IP of the instrument manufacturers. To the best of the authors' knowledge, time of flight is an equivalent term for the pulse width.

We extended the original sentence in lines 44-46 (lines 45-47 in the revised manuscript).

- Line 48: The wording in this sentence basically says the exact same thing as the previous sentence. It doesn't show that measuring these extra pulse shape characteristics can do anything extra/different than measuring pulse area, height, or width. What is basically said is that with both you can differentiate doublets, whereas the mention of pollen and other particles detracts or shows an advantage over traditional pulse shape measurements.

We apologize for imprecise wording of this statement. We clarified that the doublet detection using additional pulse shape characteristics can be more specific or sensitive than common methods, which is also supported by the cited literature. However, we disagree with the statement that measuring pulse shape characteristics does not provide a benefit over the common parameters as shown by Zilmer et al. and Godavarti et al. This is mentioned in the second part of the sentence by listing further distinction of other objects by means of pulse shape analysis. An appropriate reference is given.

We extended the sentence in lines 47-49 in the original document (lines 48-50 in the revised manuscript).

- Line 50: The literature examples here should be tied into the "pulse skewness", "kurtosis", and "frequency properties" that is mentioned previously. This would strengthen their argument that measuring additional pulse shape characteristics is useful.

We thank the reviewer for this suggestions regarding strengthen our arguments. The authors intended to illustrate the widespread applications of pulse shape analysis and therefore wanted to provide a broad spectrum of literature on these applications. We changed the sentence to separate it conceptually from the preceding sentence.

- Line 78: why is MgCl₂ used?

The authors appreciate that for RNase A MgCl₂ is not required. It was used since it is contained in buffers used for other procedures in our lab.

- Line 85: Here it seems like the cells measured only with the MAPS-FC technique are still stained. Might be good to add that this aliquot was not stained just for clarity.

The authors apologize for the incomplete description. Indeed, the cells measured with the MAPS-FC technique were stained since the staining was used as a reference to assess the performance of MAPS-FC. We clarified this point in the Methods Section of the manuscript.

A sentence was added in line 85 in the original document (lines 87-88 in the revised manuscript).

- Line 110: Owing to the limited channels usable, why chose these positions (2, 3, and 4) specifically? Why not 3, 5, and 7 or 2, 6, and 10?

The authors acknowledge this important question on the choice of the detection angles. The fiber positions were chosen for two main reasons: fiber positions along the flow direction were expected to exhibit more information due to symmetry breaking. And from the positions along the flow direction, a trade-off between very high background intensities due to the transmitted beam and low signal intensities at larger angles lead to the choice of the three fiber positions mentioned. We added a brief justification of this choice to our manuscript text.

A sentence was added to line 110 in the original document (lines 114-116 in the revised manuscript).

- Figure 2: is the laser beam smaller than the cell? So is this slit-scanning for all cell types?

In principal, the concept is similar to slit-scanning. However, we do not restrict the numerical aperture of detection but rather use a tightly focused laser beam, which realizes a situation similar to slit scanning for typical eukaryotic cells. The authors appreciate this question and added respective information with a literature reference.

The sentence in line 101 was split and an additional sentence was added (lines 104-106 in the revised manuscript).

- Figure 3: what are the upper and lower limits for transit times for the approach to work? I think a brief 1-2 sentence explanation here would be helpful to non-SME readers.

The transit time is determined by the flow speed and consequently by the fluidics pressure. With our signal processing, the acquisition time window per event, and thereby the maximum pulse length, is limited to a maximum of 16 μs. We added a brief description of these boundaries to the Methods Section. We added three sentences at line 131 in the original document (lines 136-138 in the revised manuscript).

- Line 122: It would be a good idea for the author to justify why trigger was based on SSC.

We thank the reviewer for this valuable suggestion. We now mention in the Methods Section that the SSC is chosen as the trigger channel to establish a common trigger point for all FSC channels. We extended the sentence in line 122 of the original document (lines 127-128 in the revised manuscript).

- Line 130: What are the specs for the low-pass filters used? What high-frequencies were filtered out?

We apologize for the incomplete information and added the cutoff frequency (7 MHz) of the low-pass filter

in the analog electronics.

- Line 136: How did the authors determine 80 data points? Considering this should be a highly quantitative technique, 80 data points seems quite low in terms of sampling, if this is what is meant.

The number of data points per event is given by the length of the trigger window and the sampling rate. Since a trigger window $> 8 \mu\text{s}$ was not necessary to measure the pulse length of approximately $4 \mu\text{s}$ as given by the cells under investigation, there is no need to acquire more data points. The sampling rate of 10 MHz is appropriate for the frequency components obtained in the signals as it fulfills the requirements of the Nyquist sampling theorem that the sampling rate should be 2 times the frequency of the analog signal. We clarified that a total of 640 pulse shape data points is acquired per event (8 channels times 80 samples). The sentence in line 136 of the original document was extended (lines 143-144 in the revised manuscript).

- Were the extracted wavelet coefficients details included in supplementary (in terms of conditions met since this transform will produce positive, real integers and imaginary counterparts).

The wavelet coefficients in our analyses do not contain complex numbers.

- General note: if only in supplementary, it would be better to include in the paper the explanation on how controls that discern single events from coincidence events were made.

Coincidental events were avoided by adjusting the sample concentration such that low flow rates (< 1000 evts./s) were obtained. At such flow rates, a coincidence of multiple events is highly unlikely.

- Line 147: The word channel is used a lot here and in Figure 4, however this is referring to the FSC channels and not fluorescence and SSC? Please clarify this.

The authors apologize for the confusion that may be caused by these descriptions. However, the word 'channel' indeed means all channels including SSC and fluorescence. Even though clusters were only analyzed for certain channels, they were generated for all channels. We clarified this by explicitly mentioning that all channels are included in the clustering.

The respective information was added to the sentence (line 155 in the revised manuscript).

- Line 150: With room for at least another angle measurement can this be explained if it is future work?

The authors appreciate this valuable comment. Measurements at different angles will be conducted in the future, including different cell types. We specified the respective paragraph in the Conclusion of the manuscript. Moreover, comparison of bead measurements with simulations are currently in progress. Testing of different angles was added as a future plan in the Discussion (line 328-329 in the revised manuscript).

- Results section: the first two paragraphs have numerous grammatical issues with wrong/missing commas, missing conjunctions, semicolons, etc.

We revised the Results section.

Several minor changes have been made (lines 205-234 in the revised manuscript).

- Figure 6(a): The pointer diagram are not great depictees of the data. Is there a better way to visualize clustering data?

We appreciate the external view of the reviewer on the matter and replaced the pointer diagrams with small conventional pseudocolor dotplots for each cluster.

Figure 6 was replaced by a new version. The corresponding explanations, mainly lines 202-207 in the original document, have been removed or changed (mainly lines 210-213, 216-222 in the revised manuscript).

- Line 254: Since authors do mention correlations, it would be best if they establish a hypothesis and test (t-test or similar) with an acceptable confidence interval or some other rigorous statistical approach.

The authors are convinced that the validation by experimental methods as presented in the manuscript provides a strong evidence that the results are statistically sound.

- Line 296: This might be better stated in the introduction so the reader can associate FSCL/FSCU with those phases throughout the paper.

The authors are convinced that the Introduction should be free of specific details on the experimental setup to avoid confusion and would therefore prefer to keep the definition of the abbreviations for the channels in the Methods Section.

Reviewer #2:

- While I can see the decision to validate the approach on two well characterised cell lines, my issue with this approach is always just how “real world” this is. The authors state that it is bad to label cells with dyes for DNA/cell cycle analysis, but only if you plan to try and culture them afterwards or do some function assay with them. My issue is that it feel that the problem has been “over played”.

The authors see the argument that the chosen application is only a small subset of biological research areas. However, the authors believe that for implementing the proof of principle of a new method, well-known and characterized systems are beneficial. We slightly modified the Introduction to make that more clear.

We changed lines 26-29, and 58 in the original document (lines 26-29, and 60 in the revised manuscript).

- If the authors had modified a cell sorter in this way and could select cells based on MAPS-FC then this would seem a much more convincing and impressive example of the approach.

The authors appreciate the strong interest in sorting single cells based on MAPS-FC. This will be pursued in a future project.

- It is not immediately clear why one would want to analyse cell cycle label-free or sort cells in different cell cycle stages.

We have better clarified in the introduction why label-free cell analysis and sorting is of great interest for biological research.

We changed lines 26-29 of the original document (lines 26-30 in the revised manuscript).

- The authors mention that’s scNRA seq data is often confounded by cell cycle stage but any bias in the data can be corrected by computational methods. Moreover, dyes such as Hoechst 33342 that enter live cells and label DNA can

The authors acknowledge this helpful insight. Still, the authors think that avoiding the need for corrections in the first place is preferable. Thus, label-free cell sorting would be helpful despite the availability of corrections. We slightly modified the respective part of the Introduction to weaken our original statement. We changed the statement in line 28 of the original document (lines 27-29 in the revised manuscript).

- While the optical set up in impressive, I am concerned about how it will or could be implemented in the real world and by others. As it has been built as an adaption to a commercially available system (Becton Dickinson LSRII).
 - o How would this affect service cover or maintenance schedules with the company?

To our knowledge the LSRII is not technically supported by BD anymore. In general, we cannot inform on the legal consequences for service contracts.

- o Why have the authors not build their own dedicated system?

We chose to use the existing fluidics system to concentrate our effort on the changes to the optical setup.

- o Could it be implemented in a different commercially available cytometer system?

In principal, the setup could be implemented in other commercially available instruments. The main restriction is the available space to position the required optics and detectors, which must be considered in a detailed example.

- One wonders why the authors have not tried to identify mitotic cells from G2. This would be more useful as many inhibitors of cell cycle work at prophase.

The authors agree that there is a wealth of further applications of interest to the biological research community.

- The quality of the staining for DNA (PI) seems poor with very broad peaks. Why was this?

The authors agree with the reviewer that published cell cycle analyses can have much lower CVs depending on the cells used.

- The “intensity pointer diagrams are very confusing and require better explanation with a dedicated figure or better text.

We replaced the pointer diagrams by small standard pseudocolor plots of the fluorescence intensity distribution in each cluster and hope this is less confusing for the reader. Nevertheless, the results are not changed.

Figure 6 was replaced by a new version. The corresponding explanations, mainly lines 202-207 in the original document, have been removed or changed (mainly lines 210-213, 216-222).

- I am not at all convinced by the final experiment where the authors use live, unfixed, unlabelled cells in the approach. I would like to see more data on the comparison in addition to figure 8.

The authors apologize for the low amount of data. Unfortunately, for unstained cells there is little room for providing more evidence as long as sorting of the analyzed cells based on MAPS-FC is not yet possible.

- They claim that “a flow cytometric method to identify cell cycle stages without labelling is not currently available”. This is not strictly correct, as a number of approaches have been developed using imaging flow cytometry to derive cell cycle phases without labelling with high accuracy and precision. I think it is important that the authors correct this oversight and provide some context in regard to their work. The most obvious would be to say that their approach is not image-based but still important to acknowledge that the pursuit of label-free cell cycle classification by flow cytometry is not novel. I am more puzzled as the study even uses an Imaging Flow Cytometry system.

We gratefully acknowledge the hint and apologize for the inaccuracy. We corrected this part and now briefly mention the possible drawbacks of image analysis in sorting. Moreover, we included according references.

The statement in lines 28-29 of the original document was corrected and extended (lines 29-30 in the revised manuscript).

In addition, we have specified the values for the angles in Fig. 2. according to the technical drawing.

REVIEWERS' COMMENTS:

Reviewer#1 (Comments to the Authors):

The authors have addressed the comments from the reviewers reasonably well. However, the authors have failed to mention other label-free flow cytometry methods such as quantitative phase imaging flow cytometry and Raman imaging flow cytometry. The authors should discuss them (how they are different from or similar to the authors' method) in the Introduction or Discussion. Specifically, the following label-free flow cytometry methods should be discussed:

1. Nitta et al, Nature Communications, <https://www.nature.com/articles/s41467-020-17285-3>
2. Hiramatsu et al, Science Advances, <https://advances.sciencemag.org/content/5/1/eaau0241>
3. Lee et al, Journal of Biophotonics, <https://onlinelibrary.wiley.com/doi/full/10.1002/jbio.201800479>
4. Lee et al, Trends in Biotechnology, [https://www.cell.com/trends/biotechnology/fulltext/S0167-7799\(21\)00064-0](https://www.cell.com/trends/biotechnology/fulltext/S0167-7799(21)00064-0)

If the authors address this point, I'd be happy to recommend publication of this work in Communications Biology.

Point-by-point response to the reviewers' comments

The authors greatly acknowledge the valuable feedback from the reviewer. We will address all aspects raised in the following point-by-point response.

Reviewer #1:

Reviewer#1 (Comments to the Authors):

The authors have addressed the comments from the reviewers reasonably well. However, the authors have failed to mention other label-free flow cytometry methods such as quantitative phase imaging flow cytometry and Raman imaging flow cytometry. The authors should discuss them (how they are different from or similar to the authors' method) in the Introduction or Discussion. Specifically, the following label-free flow cytometry methods should be discussed:

1. Nitta et al, Nature Communications, <https://www.nature.com/articles/s41467-020-17285-3>
2. Hiramatsu et al, Science Advances, <https://advances.sciencemag.org/content/5/1/eaau0241>
3. Lee et al, Journal of Biophotonics, <https://onlinelibrary.wiley.com/doi/full/10.1002/jbio.201800479>
4. Lee et al, Trends in Biotechnology, [https://www.cell.com/trends/biotechnology/fulltext/S0167-7799\(21\)00064-0](https://www.cell.com/trends/biotechnology/fulltext/S0167-7799(21)00064-0)

If the authors address this point, I'd be happy to recommend publication of this work in Communications Biology.

We have included the appropriate references in the Introduction and Discussion part of the MS and have addressed the relation to our work.